# Prevalence of DSM-5 mild and major neurocognitive disorder in India: Results from the LASI-DAD

**Alden L. Gross**[1,2]*, **Emma Nichols**[3], **Marco Angrisani**[3,4], **Mary Ganguli**[5], **Haomiao Jin**[6], **Pranali Khobragade**[4], **Kenneth M. Langa**[7,8,9], **Erik Meijer**[3,4], **Mathew Varghese**[10], **A. B. Dey**[11], **Jinkook Lee**[3,4]

1 Department of Epidemiology, Johns Hopkins Bloomberg School of Public Health, Baltimore, Maryland, United States of America, 2 Center on Aging and Health, Johns Hopkins University, Baltimore, Maryland, United States of America, 3 Center for Economic and Social Research, University of Southern California, Los Angeles, Los Angeles, California, United States of America, 4 Department of Economics, University of Southern California, Los Angeles, Los Angeles, California, United States of America, 5 Department of Psychiatry, University of Pittsburgh School of Medicine, Pittsburgh, Pennsylvania, United States of America, 6 School of Health Sciences, University of Surrey, Guildford, United Kingdom, 7 Department of Epidemiology, University of Michigan School of Public Health, Ann Arbor, Michigan, United States of America, 8 Department of Internal Medicine, University of Michigan Medical School, Ann Arbor, Michigan, United States of America, 9 Institute for Social Research, Veterans Affairs Center for Clinical Management Research, Ann Arbor, Michigan, United States of America, 10 Department of Psychiatry, St. John's Medical College, Bengaluru, Karnataka, India, 11 Venu Geriatric Institute, New Delhi, India

* agross14@jhu.edu

**Data Availability Statement:** The data underlying the results for LASI-DAD presented in the study are available from https://g2aging.org/?section=lasi-

## Abstract

### Introduction

India, with its rapidly aging population, faces an alarming burden of dementia. We implemented DSM-5 criteria in large-scale, nationally representative survey data in India to characterize the prevalence of mild and major Neurocognitive disorder.

### Methods

The Harmonized Diagnostic Assessment of Dementia for the Longitudinal Aging Study in India (LASI-DAD) (N = 4,096) is a nationally representative cohort study in India using multi-stage area probability sampling methods. Using neuropsychological testing and informant reports, we defined DSM-5 mild and major neurocognitive disorder, reported its prevalence, and evaluated criterion and construct validity of the algorithm using clinician-adjudicated Clinical Dementia Ratings (CDR)®.

### Results

The prevalence of mild and major neurocognitive disorder, weighted to the population, is 17.6% and 7.2%. Demographic gradients with respect to age and education conform to hypothesized patterns. Among N = 2,390 participants with a clinician-adjudicated CDR, CDR ratings and DSM-5 classification agreed for N = 2,139 (89.5%) participants.

dad-downloads. Data are shared via this website with a signed data access agreement.

**Funding:** This work was supported by the National Institute on Aging (R01 AG051125 (JL), R01 AG030153 (JL), U01 AG064948 (JL), R01 AG070953 (ALG)). The funders had no role in study design, data collection and analysis, decision to publish, or preparation of the manuscript.

**Competing interests:** The authors have no conflicts, whether personal, financial, or otherwise.

## Discussion

The prevalence of dementia in India is higher than previously recognized. These findings, coupled with a growing number of older adults in the coming decades in India, have important implications for society, public health, and families. We are aware of no previous Indian population-representative estimates of mild cognitive impairment, a group which will be increasingly important in coming years to identify for potential therapeutic treatment.

## Introduction

Indian adults aged 60 years and older number over 138 million and are projected to constitute 20% of the Indian population by 2050 [1]. With an expanding population of older adults, driven by longer life expectancy and better health care [2, 3], comes diseases and conditions common with aging. Dementia is a significant public health concern globally and especially in low and middle-income countries [4]. There are many approaches to measuring dementia that have been implemented in population-based studies, ranging from resource-intensive clinical adjudication [5, 6] to algorithms built according to *a priori* criteria [7, 8]. Algorithms based on neuropsychological testing with robust norms tend to be more stable longitudinally and are associated with fewer false positive classifications [9–11].

Previous estimates of dementia prevalence in India have been primarily based on 10/66 algorithmic criteria, which rely on cutoffs on standardized cognitive testing and evaluations of everyday functioning. According to the 10/66 algorithm, prevalence estimates range from 7.5% in urban India to 10.6% in rural India [12]. Notably, other prior studies of dementia prevalence in India have reported prevalence estimates between 2.2–14.9% but have relied on convenience samples or geographically limited samples [13, 14]. Leveraging a nationally representative sample, Lee and colleagues [15] estimated a dementia prevalence of 7.4% using a prediction algorithm informed by clinical dementia rating (CDR) [16]. The Diagnostic and Statistical Manual of Mental Disorders (DSM, 5th ed) is appealing because the DSM is a widely recognized clinical diagnostic authority [17]. Recognizing that dementia quintessentially involves cognitive decline severe enough to interfere with one's daily functioning, diagnostic criteria for DSM-5 neurocognitive disorder entails documentation of cognitive decline, assessment of independence in everyday activities, and exclusionary conditions.

There are notable distinctions between 10/66 criteria, based on DSM-IV, and DSM-5. First, the former required impairment in memory and another cognitive domain, while the latter is more inclusive in the sense that it required cognitive impairment in any domain, not just memory. Second, DSM-IV criteria required evidence that cognitive deficits led to impaired everyday functioning that represents a significant decline from a previous level. The DSM-5 revised this requirement by requiring that cognitive deficits interfere with independence in everyday activities. The requirement in DSM-IV of a decline in everyday function from a prior level was difficult to assess in older adults in India for whom assessment of independent activities of daily living can be challenging.

The present study aims to describe the prevalence of mild and major neurocognitive disorder in a nationally representative sample in India. We derived and validated a diagnostic algorithm using DSM-5 criteria. We report dementia prevalence rates overall and across demographic factors, relate self-reported memory to risk of diagnosis, and compare DSM-5 classifications to clinically adjudicated CDR scores available in a subsample of participants.

## Materials and methods

### Sample

The Longitudinal Aging Study in India (LASI) is a nationally representative sample of health, social supports, and economic security in N = 72,262 adults in India aged 45 years and older, sampled from 36 states and union territories using multistage area probability methods [18]. Part of the LASI core interview entailed a brief cognitive assessment; those unable to complete cognitive tests provided an informant interview to gauge their general mental status. The Harmonized Diagnostic Assessment of Dementia for the LASI (LASI-DAD) is a substudy comprising N = 4,096 participants [19]. The LASI-DAD study employed a stratified, random sampling design, recruiting participants from the main LASI study residing in 18 geographically and linguistically diverse states (Gujarat, Uttranchal, Jammu & Kashmir, Punjab, Haryana, Delhi, Rajasthan, Uttar Pradesh, Madhya Pradesh, Bihar, Assam, West Bengal, Orissa, Maharashtra, Karnataka, Kerala, Tamil Nadu, and Telangana), collectively representing 91% of the population [19]. LASI-DAD administered a detailed neuropsychological battery based on the Harmonized Cognitive Assessment Protocol (HCAP) developed by the closely related Health and Retirement Study in the US [20]. The HCAP collects informant interviews using self-report and standardized instruments. Informants were close family members or friends knowledgeable of the participant's everyday functioning [21]. LASI-DAD data are publicly available and available with a signed data access agreement; data for this study were accessed August 12, 2022.

LASI-DAD participants were interviewed in one of 13 languages; aside from Hindi and English, languages of administration corresponded to one's state of residence (Table 1). Cognitive tests and interview materials were carefully translated, back-translated, and rechecked for accuracy by field staff and study investigators. A previous study tested for differences in the correlational structure of the cognitive test battery by language of administration by testing configural, metric, and scalar measurement invariance, using available data on N = 3,224 participants representing 11 of the 13 languages (interviews from Gujarat and Punjab had not yet been released at that time). There was minimal evidence of measurement differences across language groups.

### Variables

To characterize cognitive performance, we leveraged summary factors representing memory, executive functioning (including abstract reasoning and attention/speed of processing), language/fluency, visuospatial ability, and orientation from a previously validated confirmatory factor analysis of the LASI-DAD neuropsychological test battery [22].

Informant-related cognitive decline was ascertained using the Informant Questionnaire on Cognitive Decline in the Elderly (IQCODE) [23], which has been previously validated in our sample [21]. The IQCODE is a screening instrument for change in cognitive and functional abilities compared to 10 years ago. Informant-rated functional decline in everyday activities was measured using Parts I (instrumental activities of daily living; IADLs) and II (activities of daily living; ADLs) of the Blessed Dementia Rating Scale [24]. Data on exclusionary conditions, including delirium, schizophrenia, and depression, were available from the core LASI interview.

LASI-DAD implemented an online clinical adjudication procedure for a subset of LASI-DAD participants [25]. Three clinicians rated each case. Clinicians were provided with informant reports, demographic information, and basic information about some cognitive testing (brief mental status tasks such as orientation to place and time and serial 7s). They were not shown results from the full battery of tests used to construct factor scores [22].

**Table 1. Demographic characteristics of the LASI-DAD sample (N = 4096).**

| Characteristic | Full sample | Robust normative sample | Not in robust normative sample |
|---|---|---|---|
| | Raw N (weighted %) or weighted mean (SD) | Raw N (weighted %) or weighted mean (SD) | Raw N (weighted %) or weighted mean (SD) |
| Sample size | 4,096 | 2,513 | 1,583 |
| Age in years, n (%) | | | |
| 60–64 | 1290 (32.1) | 873 (35.4) | 417 (26.7) |
| 65–69 | 1179 (29.6) | 759 (31.3) | 420 (26.8) |
| 70–79 | 1173 (28.8) | 687 (27.0) | 486 (31.7) |
| 80+ | 453 (9.5) | 193 (6.3) | 260 (14.7) |
| Female sex, n (%) | 2207 (50.8) | 1314 (49.4) | 893 (53.1) |
| Educational attainment, n (%) | | | |
| No formal education | 2009 (43.2) | 1182 (41.4) | 827 (46.1) |
| Secondary education or less | 1076 (29.7) | 644 (27.6) | 432 (33.2) |
| Tertiary education or higher | 1010 (27.1) | 686 (31.0) | 324 (20.7) |
| Language of test administration, n (%) | | | |
| English | 10 (0.4) | 8 (0.5) | 2 (0.1) |
| Hindi | 1393 (35.9) | 897 (36.9) | 496 (34.3) |
| Kannada | 245 (5.6) | 182 (6.7) | 63 (3.8) |
| Malayalam | 349 (8.4) | 200 (7.6) | 149 (9.6) |
| Gujarati | 288 (7.3) | 176 (6.8) | 112 (8.1) |
| Tamil | 301 (8.2) | 189 (8.7) | 112 (7.5) |
| Punjabi | 159 (4.0) | 108 (4.3) | 51 (3.3) |
| Urdu | 152 (2.1) | 97 (2.1) | 55 (2.0) |
| Bengali | 309 (7.4) | 125 (5.0) | 184 (11.4) |
| Assamese | 199 (2.9) | 126 (3.2) | 73 (2.5) |
| Odiya | 252 (4.6) | 137 (4.6) | 115 (4.8) |
| Marathi | 250 (9.3) | 144 (9.3) | 106 (9.3) |
| Telugu | 189 (4.0) | 124 (4.4) | 65 (3.4) |
| Residential status, n (%) | | | |
| Urban | 1556 (56.5) | 979 (58.4) | 577 (53.3) |
| Rural | 2539 (43.5) | 1533 (41.6) | 1006 (46.7) |
| Literacy, n (%) | | | |
| Can read or write | 1777 (49.1) | 1149 (51.2) | 628 (45.6) |
| Cannot read or write | 2319 (50.9) | 1364 (48.8) | 955 (54.4) |
| Factor scores for cognitive function, mean (observed range) | | | |
| Memory | 0.1 (-2.2, 3.7) | 0.2 (-2.2, 3.7) | -0.1 (-2.2, 3.3) |
| Executive functioning | 0.1 (-1.9, 2.5) | 0.2 (-1.9, 2.5) | -0.1 (-1.9, 2.5) |
| Language | 0.1 (-3.4, 2.0) | 0.2 (-3.4, 2.0) | -0.1 (-3.4, 2.0) |
| Visuospatial | 0.1 (-1.6, 1.6) | 0.2 (-1.6, 1.6) | -0.1 (-1.6, 1.6) |
| Orientation | 0.1 (-2.5, 0.9) | 0.2 (-2.5, 0.9) | -0.1 (-2.5, 0.9) |
| Jorm IQCODE, mean (SD) | 3.4 (0.0) | 3.3 (0.0) | 3.5 (0.0) |
| Impairment in any IADL, n (%) | | | |
| No loss | 2570 (63.9) | 1678 (67.9) | 892 (57.4) |
| Some loss | 962 (23.8) | 568 (22.7) | 394 (25.4) |
| Severe loss | 564 (12.3) | 267 (9.4) | 297 (17.2) |
| Impairment in any ADL, n (%) | 384 (8.5) | 132 (5.0) | 252 (14.3) |

Raw sample sizes are provided with weighted means and proportions. ADL: activities of daily living; IADL: instrumental activities of daily living; SD: standard deviation; IQCODE: Informant Questionnaire on Cognitive Decline in the. Elderly.

## Analysis

We characterized sample demographics. Next, we operationalized an algorithm for DSM-5 major and mild neurocognitive disorder and evaluated its criterion-related validity. Table 2 summarizes the algorithm. The four major components were objective cognitive decline, informant-rated cognitive decline, informant-rated functional decline, and exclusionary conditions.

**Criteria for objective cognitive decline.**   To operationalize objective cognitive decline in this cross-sectional sample, criteria for cognitive impairment were identified using a robust normative sample in which we calculated cutoffs on cognitive performance. The four cognitive domains we used were memory, executive functioning, language/fluency, and visuospatial ability [22], pursuant to the DSM-5 [17]. Criteria for major neurocognitive disorder require either (a) impairment of 1.5 standard deviations (SD) or worse in two or more domains [26], or (b) impairment of 1.5SD in one domain alongside impairment of 1SD or worse in at least two other domains. Criteria for mild neurocognitive disorder required impairment of 1SD in at least two domains.

We identified a robust normative sample of LASI-DAD participants using health, functional, and demographic information from the core LASI interview. We excluded participants whose LASI core interview included a proxy interview (N = 62) or who had a history of stroke (N = 289), major depressive disorder according to the Composite International Diagnostic Interview (CIDI) (N = 307) [27], intellectual or learning difficulty (N = 76), family history of Alzheimer's or Parkinson's disease (N = 137), or self-reports of any ADL difficulty (bathing, dressing, toileting, transferring, eating, incontinence) (N = 712). Within this normative group of N = 2,513 (61.4%), we identified cutoffs of 1 SD and 1.5 SD below the mean on each cognitive domain score adjusted for age, gender, literacy, education, and all two-way interactions among these variables.

**Criteria for informant-rated cognitive decline.**   Informant-rated cognitive decline was operationalized using the IQCODE [23]. Based on prior research [23, 26], we used a cutoff of ≥3.5 for major neurocognitive disorder and ≥3.2 for mild neurocognitive disorder [28].

**Criteria for functional decline in everyday activities.**   The criterion for major neurocognitive disorder was met if informants reported any ADL difficulty (Blessed Part 2 score ≥1), or any severe IADL loss (Blessed Part 1 score ≥2). The criterion for mild neurocognitive disorder was met if participants needed no assistance with any ADL and had either no or some loss in any IADL. Criteria for mild neurocognitive disorder was also met if ADL impairment and

**Table 2. Operationalization of DSM-5 criteria for major and mild neurocognitive disorder: Results from LASI-DAD (N = 4096).**

| DSM5 criteria | Major NCD | Mild NCD |
|---|---|---|
| A1. Informant-rated cognitive decline | IQCODE ≥3.5 points and poor self-rated memory | IQCODE ≥3.2 points or poor self-rated memory |
| A2. Objective cognitive decline | (a) Impairment of 1.5SD or worse in 2+ domains <or> (b) Impairment of 1.5SD or worse in 1 domain + impairment of 1SD or worse in 2+ other domains | impairment of 1SD or worse in 1 + domain |
| B. Functional decline in everyday activities | Any ADL impairment, Blessed Part 1 (IADL) score 2 or greater, or Blessed Part 2 (ADL) score greater than 1 | (a) No ADL impairment and no/ minimal loss in IADLs <or> (b) Discordant informant reports for ADL vs IADL impairment |
| C/D. Exclusionary conditions | No schizophrenia, active delirium during testing, or history of major depression | |

IADL difficulty were discordant; such discordant reporting suggests the informant is reporting an isolated issue, or a disagreement that is not clinically significant.

**Exclusionary conditions.** Exclusionary conditions that would preclude a neurocognitive disorder classification included active delirium, schizophrenia, history of stroke, and major depression, all of which were ascertained using data from the core LASI interview.

**Validation.** We evaluated criterion and construct validity of our measure of DSM-5 major and mild neurocognitive disorder in three ways. First, we compared patterns of prevalence estimates by age, educational attainment, gender, literacy level, and urbanicity. Second, we tested the correspondence between the measure and self-reports of two-year change in memory using logistic regression adjusted for age, educational attainment, gender, literacy, and urbanicity. Third, we cross-tabulated DSM-5 classifications against CDR scores. Because raters sometimes assigned disparate CDR scores which were resolved through consensus, to present a clear comparison to DSM-5, we excluded N = 137 cases for which there was a major inter-rater inconsistency (CDR rating ≤0.5, vs ≥1). Thus, we compared DSM-5 criteria with CDR scores in N = 2390 cases in which there were no major disagreements among three clinicians. We assessed correspondence between our measure and the six CDR subscales [16].

**Sensitivity analysis.** DSM-5 criteria for major and mild neurocognitive disorder list cognitive domains to be considered, including memory, complex attention and executive functioning, language, and visuospatial ability [17]. Orientation is not a suggested domain because it is more an immediate assessment of one's awareness and understanding of surroundings. That said, consistent with dementia algorithm research in the US, in a sensitivity analysis we added a fifth factor score for orientation [26].

Analyses were conducted using Stata 17 [29]. We report unweighted and weighted prevalence estimates using population weights scaled to the population of adults 60 years and over in India [30].

## Results

The LASI-DAD sample is aged between 60 and 104 years, with a majority between 60 and 79 years of age (mean 69 years) (Table 1). The weighted proportion of women is 51%, while 43% report no formal education and most (56%) live in an urban setting. In the full sample, 12% reported severe loss in at least one ADL and 8.5% reported impairment in any IADL. Those selected into the robust normative sample (N = 2,513) were more likely to be younger, more educated, literate, and urban. Importantly, while mean cognitive scores in the robust normative subgroup were higher on average than those not in the subgroup, the range of cognitive scores overlapped greatly. Underscoring the assumption that discordant information between informant-reported IADL and ADL difficulty is likely not clinically significant, younger-generation informants (e.g., children who are less likely to live with the participant) are 1.3 times more likely (95% confidence interval 1.1, 1.6) to report such discordant patterns on the Blessed than same-generation informants (e.g., spouses).

### Overall prevalence

The unweighted prevalence of DSM-5 major and mild neurocognitive disorder in the sample is 8.5% (N = 348) and 17.4% (N = 712), respectively. Applying the sampling weights, the respective estimates of population prevalence are 7.2% and 17.6%.

### Prevalence in subgroups

The weighted prevalence of major neurocognitive disorder is greater with older ages (ranging from 3.8% among those aged 60–64 years to 15.2% among those aged 80+ years) and at lower

**Table 3. Prevalence of major and mild neurocognitive disorder by demographic characteristics: Results from LASI-DAD (N = 4096).**

| Characteristic | Sample size (unweighted) | Mild NCD, weighted % | Major NCD, weighted % |
|---|---|---|---|
| Full sample | 4096 | 17.6 | 7.2 |
| Age | | | |
| 60–64 | 1290 | 14.7 | 3.8 |
| 65–69 | 1179 | 17.1 | 6.7 |
| 70–79 | 720 | 19.2 | 8.9 |
| 80+ | 453 | 23.9 | 15.2 |
| Cumulative age groups | | | |
| 65+ years | 2806 | 18.9 | 8.8 |
| 70+ years | 1627 | 20.4 | 10.5 |
| Education | | | |
| No formal education | 2009 | 18.9 | 10.8 |
| Secondary education or less | 1076 | 17.8 | 5.4 |
| Tertiary education or higher | 1010 | 15.2 | 3.5 |
| Sex | | | |
| Male | 1889 | 16.6 | 8.4 |
| Female | 2207 | 18.5 | 6.1 |
| Literacy | | | |
| Literate | 1777 | 17.9 | 5.0 |
| Illiterate | 2319 | 17.3 | 9.3 |
| Residential status | | | |
| Urban | 1556 | 16.0 | 4.9 |
| Rural | 2539 | 19.7 | 10.3 |

Raw sample sizes are provided with weighted proportions.

levels of education (ranging from 10.8% among those with no formal education to 3.5% among those with a tertiary or higher education) (Table 3). As with major neurocognitive disorder, the prevalence of mild neurocognitive disorder was lowest in the youngest age group, and higher with older ages. The prevalence of major neurocognitive disorder was comparable between women (6.1%) and men (8.4%), as was mild neurocognitive disorder (18.5% for women vs 16.6% for men). Major neurocognitive disorder was more prevalent among illiterate (9.3%) than literate (5.0%) and rural (10.3%) than urban (4.9%) participants.

## Comparison of DSM-5 classifications with self-rated memory

Compared to those who reported their memory was the same or better compared to two years ago (N = 1,181), those reporting worse memory (N = 1,302) had 1.4 times greater odds of mild neurocognitive disorder (95% CI: 1.2, 1.7) and 4.5 times greater odds of major neurocognitive disorder (95% CI: 3.6, 5.8).

## Comparison of DSM-5 classifications with CDR ratings

Among participants with a CDR score, 89.5% had a DSM-5 classification corresponding to a comparable CDR score (e.g., CDR $\leq$0.5 corresponding to DSM-5 normal; CDR 1 to mild neurocognitive disorder; CDR $\geq$1 to major neurocognitive disorder) (Table 4). From the online adjudication procedure for the CDR, N = 1,491 (62.4%) participants were assigned a value of 0.5, indicating either mild cognitive impairment or questionable cases due to insufficient or

**Table 4. Cross-tabulation of CDR scores with DSM-5 algorithmic neurocognitive disorder: Results from LASI-DAD (N = 2390).**

| CDR score | Algorithmic neurocognitive disorder, n | | |
|---|---|---|---|
| | Cognitively normal | Mild neurocognitive disorder | Major neurocognitive disorder |
| 0 | 718 | 41 | 5 |
| 0.5 | 1,012 | 319 | 160 |
| 1 | 23 | 20 | 64 |
| 2 | 2 | 0 | 21 |
| 3 | 0 | 0 | 5 |

conflicting information available to clinicians. Among the N = 135 participants with CDR≥1, N = 20 (14.8%) and N = 90 (66.7%) were classified with mild or major neurocognitive disorder, respectively. Among participants with a CDR ≥2, 93% were classified as major neurocognitive disorder. Treating CDR≥1 as a gold standard, the DSM-5 classification has 0.66 sensitivity and 0.93 specificity for classifying dementia (Area under the curve = 0.80). The high specificity of the DSM-5 classification indicates there are few false positives which would typically falsely inflate the prevalence. Thus, the reported prevalence is unlikely to be an overestimate insofar as clinical dementia ratings can be accepted as a gold standard. The sensitivity of the DSM-5 classification is 0.66, indicating a sizable number of people with dementia via the CDR but without major neurocognitive disorder via DSM-5; of these N = 45 participants, 20 (44%) were classified with mild neurocognitive disorder and the rest (N = 25) were mildly impaired based on informant criteria but had minimal to no evidence of objective cognitive impairment (Table 2).

Among N = 255 participants with a CDR score and DSM-5 major neurocognitive disorder, 165 (64.7%) had a CDR score of ≤0.5. To better understand discrepancies between CDR ratings and the DSM-5 algorithm, Table 5 presents an analysis involving CDR subscales. Among N = 165 participants unimpaired according to CDR and who had major neurocognitive disorder, clinician raters had disagreements on subscale scores for 11–46% of participants. In the judgment and memory subscales, ratings of ≥1 were present among 49–56% of these participants, respectively. Compared to others with CDR ≤0.5, this group tended to be older (p<0.01) and more rural (p = 0.003) but did not differ by gender, literacy, or education. Compared to others with major neurocognitive disorder, this group tended to be younger (p = 0.001) and more literate (p = 0.03).

**Table 5. Agreement between DSM-5 neurocognitive disorder Algorithm and CDR ratings: Results from LASI-DAD (N = 2390).**

| CDR subdomain | Discordant cases: CDR-normal (0,0.5) but DSM5 major NCD | | Concordant cases: CDR-dementia (1,2,3) and DSM5 major NCD | | Concordant cases: CDR-normal (0,0.5) and DSM5 no or mild NCD | |
|---|---|---|---|---|---|---|
| | N = 165 | | N = 90 | | N = 2,090 | |
| | Disagreements, Number (%) | Has a CDR subscale rating of (1,2,3), Number (%) | Disagreements, Number (%) | Has a CDR subscale rating of (1,2,3), Number (%) | Disagreements, Number (%) | Has a CDR subscale rating of (1,2,3), Number (%) |
| Judgment and problem-solving | 76 (46.1) | 92 (55.8) | 17 (18.9) | 90 (100) | 358 (17.1) | 387 (18.5) |
| Memory | 65 (39.4) | 80 (48.5) | 15 (16.7) | 87 (96.7) | 347 (16.6) | 374 (17.9) |
| Orientation | 37 (22.4) | 47 (28.5) | 20 (22.2) | 87 (96.7) | 186 (8.9) | 199 (9.5) |
| Community affairs | 30 (18.2) | 34 (20.6) | 22 (24.4) | 84 (93.3) | 142 (6.8) | 148 (7.1) |
| Home and hobbies | 36 (21.8) | 38 (23.0) | 15 (16.7) | 89 (98.9) | 128 (6.1) | 135 (6.5) |
| Personal care | 19 (11.5) | 29 (17.6) | 8 (8.9) | 56 (62.2) | 71 (3.4) | 93 (4.4) |

CDR: Clinical Dementia Rating

In contrast to this discordant group, the two concordant groups in Table 5 had fewer disagreements among raters, and CDR subscale ratings were more likely to align with the assigned global CDR. Among the N = 90 participants for whom the overall CDR rating indicated dementia and were classified with DSM-5 major neurocognitive disorder, disagreements among clinician raters were low across CDR subscales (9–19%). Likewise, among the N = 2,090 participants without dementia based on CDR and no or mild DSM-5 neurocognitive disorder, clinicians disagreed on CDR subscale ratings in only 3–17% of participants.

**Sensitivity analysis.** Inclusion of orientation as a fifth domain yielded slightly higher prevalence estimates of mild (18.6%) and major (7.8%) NCD, but patterns of prevalence by demographic characteristics were unchanged (S1 Table). Correspondence with CDR scores, for which orientation items were available to clinicians, was 89.0% which is comparable to results that do not use orientation (S2 Table).

## Discussion

We determined prevalence of DSM-5 mild and major neurocognitive disorder using data from a nationally representative sample of adults in India 60 years of age and older. Given an estimated 138 million adults over 60 years of age in India, these estimates suggest approximately 24 million and 9.9 million older adults in India are living with mild and major neurocognitive disorder, respectively. Prevalence is higher with older age, less educational attainment, and among illiterate and rural-living older adults. DSM-5 criteria are broadly concordant with clinician-adjudicated CDR. These findings highlight the growing importance of dementia in India.

Our findings are similar to recent estimates of dementia prevalence in India [12, 15]. We are not aware of previous prevalence estimates for mild cognitive impairment in India; our weighted estimate of 18.9% among those ≥65 years of age is comparable to 22% in the US [26]. Especially given recent therapeutic advances in treatments for Alzheimer's disease, in coming years there may be increased emphasis on defining and identifying those with preclinical dementia and milder impairments (e.g., mild neurocognitive disorder) [31].

An innovation of this population-based study is that we operationalized DSM-5 criteria using survey information. There are many criteria for dementia ascertainment including NINCDS/ADRDA [32], comprehensive neuropsychological criteria [9], and others. While most criteria require similar components (e.g., documented cognitive decline or impairment, interference in everyday activities, exclusionary conditions), how components are operationalized and combined can lead to differences in prevalence and correlates with risk factors. Erkunjunti [33] examined agreement among six dementia classification criteria, finding the prevalence of dementia ranged from 3.1% using ICD-10 to 29.1% using DSM-III. Another study evaluated reasons for discrepancies in prevalence of Alzheimer's disease (AD) in the US between the Aging, Demographics, and Memory Study substudy of the Health and Retirement Study and the Chicago Health and Aging Project, where numbers of adults with AD in the US was estimated to be 2.3 million and 4.5 million, respectively [34]. Key reasons for that discrepancy included different pathological attributions of dementia to AD vs vascular or mixed pathologies, and differential requirements regarding informant-rated functional limitations. Requiring informant ratings of impairment in addition to objective cognitive testing, as was done in the LASI-DAD study, may identify fewer, more severe dementia cases than if only objective cognitive testing is required [34].

We did not find a strong sex difference in the prevalence of neurocognitive disorder. Among women, the prevalence of mild neurocognitive disorder was slightly greater while the prevalence of major neurocognitive disorder was slightly lower, but not substantially so. While

this pattern of results is consistent with several recent studies of dementia prevalence in India [35–38], other studies report higher dementia prevalence among women [15] and especially among women in rural areas [12], possibly due to higher risk factor profiles such as lower education, illiteracy, greater exposure to indoor pollution from cooking, and vascular risk factors [39, 40]. Data from high-income countries in Europe and North America are mixed in this regard; a recent study suggested that higher prevalence estimates of dementia among women, may be attributable to longer life expectancy among women as well as selection effects related to clinical versus population-based samples [39, 41].

Because orientation reflects an assessment of a person's general understanding of surroundings rather than the ability to remember, think, or attend to stimuli, orientation is not a domain listed in DSM-5. We kept it out of our algorithm for this reason and also to not contaminate the validation of our DSM-5 algorithm against CDR ratings. Orientation items were available to clinicians who rated CDRs using the online adjudication platform. That said, algorithms that include criteria for impairment in 1+ domains or 2+ domains such as ours, the total number of possible domains matters. We used four domains (memory, executive functioning, language, and visuospatial ability), while another algorithm using HCAP data from the US used those domains as well as orientation [26].

Future studies should leverage forthcoming longitudinal data from the LASI-DAD sample to capture cognitive decline (a second LASI-DAD wave is currently underway). In lieu of evidence of longitudinal decline, we leveraged a robust normative approach to identify lower-than-expected performance in a robust normative group that was designed to be relatively free of pathological cognitive impairment. Importantly, there was strong overlap in observed cognitive performance between those who were and were not in the robust normative sample, suggesting participants with low cognitive function, but no evidence of pathological impairment, were represented in the robust normative sample.

We used epidemiologic data to inform an algorithm to operationalize DSM-5 criteria in LASI-DAD. This classification should not be confused with clinical diagnosis. Diagnoses are generally made by a trained clinician via in-person clinic visits who can probe patients and informants for additional clarifications, or via adjudication panels of clinicians and neuropsychologists or neurologists who review available information. Classifications, such as those in this study, should not be used for individual-level decision-making, but can be used to make population-based inferences and to guide policy.

Comparison of our DSM-5 algorithm to clinician-rated CDR scores revealed strong agreement. There were discordant cases (N = 165, 6.9%) in which CDR scores suggested minimal cognitive impairment but DSM-5 suggested major impairment. Within this discordant group, we found disagreements in subscale ratings among clinical raters–especially in CDR subdomains of judgment and memory. This pattern suggests these discordant cases may have been challenging for clinical adjudicators using online procedures.

We leveraged carefully collected epidemiologic data and validated cognitive factors from a rich battery of cognitive tests. A major strength is the study's representative sampling design and population weights that enable us to make inferences about the population of India aged 60 years and older. An important limitation is incomplete life history information to inform exclusionary conditions and cognitive change. The exclusions we considered, namely schizophrenia and delirium during an interview, excluded no participants from a classification of neurocognitive disorder, likely because the probability of recruiting and interviewing such participants is low. Another limitation is that our prevalence estimates and more broadly our ability to apply algorithms to available data depends on the quality of the data; it is important to identify informants who have familiarity with a person's living situation. A third limitation is that our study lacks a clear gold standard with which to gauge the classification quality of our

algorithm. We used several independent external criteria that are known to covary with dementia, including age and education, and compared our classifications with clinician-adjudicated CDR scores, which provided strong agreement.

We used data from a population survey in India to classify participants by DSM-5 major and mild neurocognitive disorder criteria. The considerable prevalence of mild and major neurocognitive disorder we reported, coupled with a growing number of older adults in the coming decades in India, has important implications for society, public health, and families.

## Supporting information

**S1 Table. Prevalence of major and mild neurocognitive disorder by demographic characteristics using orientation as a cognitive domain to define neurocognitive disorder: Results from LASI-DAD (N = 4096).**
(DOCX)

**S2 Table. Cross-tabulation of CDR scores with DSM-5 algorithmic neurocognitive disorder using orientation as a domain to define neurocognitive disorder: Results from LASI--DAD (N = 2390).**
(DOCX)

## Acknowledgments

We are grateful to participants of the LASI-DAD study for their time and contributions to this study. All participants provided informed consent for this study. Consent for cognitively impaired participants was taken from a close family member (e.g., adult child or spouse) who could legally represent the participant. Interviewers read the consent to participants unable to read the consent. A thumb impression was accepted in place of a signature for participants unable to sign consent forms. Consent forms were translated into local languages. Ethical approval was obtained by the University of Southern California Institutional Review Board (UP15-00684) and the Indian Council of Medical Research for the All India Institute of Medical Science (54/01/Indo-foreign/Ger/16-NCD-II).

## Author Contributions

**Conceptualization:** Alden L. Gross, Emma Nichols, Mary Ganguli, Kenneth M. Langa, Erik Meijer, Jinkook Lee.

**Data curation:** Alden L. Gross, Emma Nichols, Jinkook Lee.

**Formal analysis:** Alden L. Gross.

**Funding acquisition:** Kenneth M. Langa, Mathew Varghese, A. B. Dey, Jinkook Lee.

**Methodology:** Alden L. Gross, Emma Nichols, Marco Angrisani, Haomiao Jin, Erik Meijer.

**Project administration:** A. B. Dey, Jinkook Lee.

**Validation:** Alden L. Gross.

**Writing – original draft:** Alden L. Gross, Mathew Varghese.

**Writing – review & editing:** Alden L. Gross, Emma Nichols, Marco Angrisani, Mary Ganguli, Haomiao Jin, Pranali Khobragade, Kenneth M. Langa, Erik Meijer, A. B. Dey, Jinkook Lee.

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
