## [Decision Letter · Decision Letter 0]

25 Sep 2023

PONE-D-23-26576Prevalence of DSM-5 mild and major neurocognitive disorder in India: Results from the LASI-DADPLOS ONE

Dear Dr. Gross,

Thank you for submitting your manuscript to PLOS ONE. After careful consideration, we feel that it has merit but does not fully meet PLOS ONE’s publication criteria as it currently stands. Therefore, we invite you to submit a revised version of the manuscript that addresses the points raised during the review process.

Please submit your revised manuscript by Nov 09 2023 11:59PM If you will need more time than this to complete your revisions, please reply to this message or contact the journal office at plosone@plos.org. Please include the following items when submitting your revised manuscript:A rebuttal letter that responds to each point raised by the academic editor and reviewer(s). You should upload this letter as a separate file labeled 'Response to Reviewers'.A marked-up copy of your manuscript that highlights changes made to the original version. You should upload this as a separate file labeled 'Revised Manuscript with Track Changes'.An unmarked version of your revised paper without tracked changes. You should upload this as a separate file labeled 'Manuscript'.If applicable, we recommend that you deposit your laboratory protocols in protocols.io to enhance the reproducibility of your results. Protocols.io assigns your protocol its own identifier (DOI) so that it can be cited independently in the future. For instructions see: https://journals.plos.org/plosone/s/submission-guidelines#loc-laboratory-protocols. Additionally, PLOS ONE offers an option for publishing peer-reviewed Lab Protocol articles, which describe protocols hosted on protocols.io. Read more information on sharing protocols at https://plos.org/protocols?utm_medium=editorial-email&utm_source=authorletters&utm_campaign=protocols.

We look forward to receiving your revised manuscript.

Kind regards,

Amina Nasri

Academic Editor

PLOS ONE

Journal Requirements:

2. Please include a copy of Table 5 which you refer to in your text on page 11 and 12.

3. Please upload a copy of Supporting Information Figure/Table/etc. Supplemental table 1 and 2 which you refer to in your text on page 12.

Reviewers' comments:

Reviewer's Responses to Questions

**Comments to the Author**

1. Is the manuscript technically sound, and do the data support the conclusions?

Reviewer #1: Yes

Reviewer #2: Yes

2. Has the statistical analysis been performed appropriately and rigorously? 

Reviewer #1: Yes

Reviewer #2: Yes

3. Have the authors made all data underlying the findings in their manuscript fully available?

Reviewer #1: Yes

Reviewer #2: Yes

4. Is the manuscript presented in an intelligible fashion and written in standard English?

Reviewer #1: Yes

Reviewer #2: Yes

5. Review Comments to the Author

Reviewer #1: The authors used data from a nationally representative sample to determine the prevalence of neurocognitive disorders in India. The used the DSM-5 criteria to construct the variable of interest. The topic is extremely relevant and the manuscript is well-written and informative. I have a few concerns:

1) Equating neurocognitive disorder as constructed here with MCI and AD may have some issues. What implications do the sensitivity and specificity of the diagnostic construct used here have on the final prevalence estimates?

2) The discussion can include something brief on the sensitivity analysis - why was it considered and what are the implications of the result of the sensitivity analysis?

3) The article is silent on the linguistic diversity of India. How many language versions were used? Would linguistic diversity affect the estimates or not?

Reviewer #2: 1. Number of women in India reporting to dementia clinics/memory clinics is significantly less compared to men; and this gender bias has been reported widely. Please clarify this based on your data.

2. Please specify clearly where and how the sample were collected.

3. Authors are concluding that the data reported in the study has higher prevalence than previously reported data; however, line 84 & 85 suggests that the participants and informants were interviewed both from home and hospital. Hence this claim looks superfluous.

6. PLOS authors have the option to publish the peer review history of their article (what does this mean?). If published, this will include your full peer review and any attached files.

Reviewer #1: No

Reviewer #2: **Yes: **Dr Vikas Dhikav, Medical Scientist-E, ICMR-DHR, MoHFW, Govt of India, New Delhi

---

## [Author Response · Author response to Decision Letter 0]

7 Oct 2023

Please refer to the response letter that we uploaded as an MS Word document as part of this submission.

---

## [Decision Letter · Decision Letter 1]

2 Jan 2024

Prevalence of DSM-5 mild and major neurocognitive disorder in India: Results from the LASI-DAD

PONE-D-23-26576R1

Dear Dr. Gross,

We’re pleased to inform you that your manuscript has been judged scientifically suitable for publication and will be formally accepted for publication once it meets all outstanding technical requirements.

Kind regards,

Amina Nasri

Academic Editor

PLOS ONE

Additional Editor Comments (optional):

Reviewers' comments:

Reviewer's Responses to Questions

**Comments to the Author**

1. If the authors have adequately addressed your comments raised in a previous round of review and you feel that this manuscript is now acceptable for publication, you may indicate that here to bypass the “Comments to the Author” section, enter your conflict of interest statement in the “Confidential to Editor” section, and submit your "Accept" recommendation.

Reviewer #1: All comments have been addressed

Reviewer #2: All comments have been addressed

2. Is the manuscript technically sound, and do the data support the conclusions?

Reviewer #1: Yes

Reviewer #2: Yes

3. Has the statistical analysis been performed appropriately and rigorously? 

Reviewer #1: Yes

Reviewer #2: Yes

4. Have the authors made all data underlying the findings in their manuscript fully available?

Reviewer #1: Yes

Reviewer #2: Yes

5. Is the manuscript presented in an intelligible fashion and written in standard English?

Reviewer #1: Yes

Reviewer #2: Yes

6. Review Comments to the Author

Reviewer #1: I commend the authors' efforts to incorporate the suggestions made for improving the manuscript. The manuscript is a useful addition to knowledge in this area.

Reviewer #2: Looks much better. The response to the comments have been addressed. Gender bias in reporting dementia is an important aspect of seeking dementia care in India reported in few studies.

This may be cause of a number of factors e.g. perceived importance of seeking care, gender issues prevalent in society and may be cause of lack of awareness about dementia care for women in general.

The last point however pertains to both. Still, this is an important aspect of dementia/memory clinic service utilisation which has been noticed by several clinician colleagues.

Authors have clarified the issue related to sampling, its process and the way the conclusions have been drawn which has improved the clarity of the manuscript overall.

Authors have clarified that participants were interviewed in a hospital

setting if they lived near a participating hospital and preferred to be interviewed there

rather than at home. Though authors state that they do not believe the location of interview would affect their results- to avoid confusion for readers, authors have revised the manuscript to remove the sentence.

7. PLOS authors have the option to publish the peer review history of their article (what does this mean?). If published, this will include your full peer review and any attached files.

Reviewer #1: **Yes: **Ravi Prasad Varma

Reviewer #2: **Yes: **Vikas Dhikav, Scientist-E (Medical), ICMR-Department of Health Research, MoHFW, Govt of India

---

## [Editor Report · Acceptance letter]

12 Jan 2024

PONE-D-23-26576R1 

PLOS ONE

Dear Dr. Gross, 

I'm pleased to inform you that your manuscript has been deemed suitable for publication in PLOS ONE. Congratulations! Your manuscript is now being handed over to our production team.

Kind regards, 

on behalf of

Dr. Amina Nasri 

Academic Editor

PLOS ONE